

# A multi-decadal wind-wave hindcast for the North Sea 1949 – 2014: coastDat2

Nikolaus Groll[1] and Ralf Weisse[1]

[1]Institute for Coastal Research, Helmholtz-Zentrum Geesthacht, 21502 Geesthacht, Germany

*Correspondence to:* Nikolaus Groll (nikolaus.groll@hzg.de)

**Abstract.**

Long and consistent wave data are important for analysing wave climate variability and change. Moreover, such statistics are also needed in coastal and offshore design and for addressing safety-related issues at sea. Using the third-generation spectral wave model WAM a multi-decadal wind-wave hindcast for the North Sea covering the period 1949–2014 was produced. The hindcast is part of the coastDat database representing a consistent and homogenous met-ocean data set. It is shown that despite not being perfect, data from the wave hindcast are generally suitable for wave climate analysis. In particular comparisons of hindcast data with in situ and satellite observations show on average a reasonable agreement while a tendency towards overestimation of the highest waves could be inferred. Despite these limitations, the wave hindcast still provides useful data for assessing wave climate variability and change as well as for risk analysis, in particular when conservative estimates are needed. Hindcast data are stored at the World Data Center for Climate (WDCC) and can be freely accessed using the doi:10.1594/WDCC/coastDat-2_WAM-North_Sea (Groll and Weisse, 2016) or via the coastDat web-page http://www.coastdat.de.

## 1 Introduction

Multi-decadal wind-wave hindcasts have become a common tool in supporting the assessment of wave climate variability and change, such as its extremes, trends or seasonal and inter-annual to decadal variability (e.g. WASA-Group, 1998; Sterl et al., 1998; Cox and Swail, 2001; Weisse and Günther, 2007; Dodet et al., 2010). Data from wind-wave hindcast data are also frequently used in practically oriented applications such as in navigation, shipbuilding, offshore design or strategic planning of logistics for the operation of future offshore wind farms (Weisse et al., 2009, 2015). Further applications comprise studies such as evaluating the impact of waves on sea salt emissions (Neumann et al., 2016) or the evaluation of the potential success or failure of different response strategies to oil pollution (Schwichtenberg et al., 2016). For all these different types of applications long, homogeneous and consistent wind-wave data are needed to derive robust estimates of wind-wave related parameters specific to the problem. Often such information is unavailable from in situ or satellite data alone and multi-decadal wind-wave hindcasts have become a common approach in complementing such analyses.

While formerly wind-wave hindcasts were developed independently using atmospheric forcing available at that time, they nowadays form an integral part of global atmospheric reanalysis systems (e.g. Dee et al., 2011; Chawla et al., 2013). Owing to



the limited spatial and temporal resolution of such global reanalyses, there is, however, still a substantial number of regional efforts aiming at higher spatial and temporal resolution. These efforts usually use the traditional approach in which downscaled global reanalysis wind fields are subsequently used to run a wave hindcast system over the reanalysis period or sub-periods within that period (e.g. Charles et al., 2012; Reguero et al., 2012, 2013; Bertin et al., 2013; Ponce de León and Guedes Soares,

2015).

For the North Sea, there presently exist a number of wind-wave hindcasts covering at least some decades of years. The first of these multi-decadal hindcasts was, to our knowledge, developed within the WASA project (WASA-Group, 1998) while a more recent approach based on the ERA-40 global atmospheric reanalysis (Uppala et al., 2005) is described in Reistad et al. (2011). Aiming at a consistent description of met-ocean conditions in the North Sea Weisse and Günther (2007) developed

and described a multi-decadal wind-wave hindcast for the southern North Sea that is part of a comprehensive met-ocean downscaling known as coastDat1 (Weisse et al., 2009). The latter is based on the global NCEP/NCAR reanalysis (Kalnay et al., 1996) and provides a consistent met-ocean data set comprising of regionally downscaled atmospheric (Feser et al., 2001), tide-surge (Weisse and Plüss, 2006) and wind-wave Weisse and Günther (2007) conditions from which full met-ocean data are available for every hour over the hindcast period.

Data from coastDat1 were extensively used for a wide variety of studies. For an overview see for example Weisse et al. (2009). While there was and there still is substantial interest in these data, the effort terminated in 2007 when the atmospheric component of the met-ocean hindcast was discontinued and consistent atmospheric data to drive the wind-wave and tide-surge models became unavailable. Up to that time, data from coastDat1 have been used by more than 50 different users with a large variety of applications. About 50 % of these users originated from commercial enterprises, while about 25 % had a more direct

scientific interest and another 25 % came from public authorities (Weisse et al., 2015). Because of ongoing interest in both, the scientific and the commercial exploitation of the data, eventually a following-up effort called coastDat2 was initiated. In this effort, upgraded models with higher spatial resolution were used and the wind-wave part, in addition, now also covered the entire North Sea. The atmospheric component of coastDat2 is described in Geyer (2014). In the following, we describe and evaluate the upgraded wind-wave part (Groll and Weisse, 2016) that is driven by the coastDat2 wind fields (Geyer, 2014).

## 25   2   Model setup

For the wind-wave hindcast in coastDat2 the third-generation spectral wind-wave model WAM (WAMDI-Group, 1988; Komen et al., 1996) version 4.5.4 was used. This version represents an update and extension of the WAM cycle 4 used in coastDat1 (Weisse and Günther, 2007) the details of which are described in the WAM documentation available at http://mywave.github. io/WAM/. Apart from some technical changes such as in the I/O routines, a major change was introduced by the replacement

of the wave dissipation source term with a new version described in Bidlot et al. (2005) and Bidlot et al. (2007).

For the coastDat2 hindcast, the wave model was used in a nested version with a coarse grid simulation covering most of the northeast North Atlantic and a fine grid simulation covering the North Sea from 4.75°W –13.25°E and from 50.5°N –59.5°N





(Figure 1). The grid size of the coarse grid is 0.5°latitude x 0.75°longitude while that of the fine grid nested within the coarse grid is 0.05°latitude x 0.075°longitude. The latter corresponds to a grid spacing of approximately 3-by-3 nautical miles.

Wave spectra were computed and discretised using 35 frequencies ranging from approx. 0.042 Hz to 1.067 Hz and 24 directional bins. The integration time step was set to three minutes and integrated parameters derived from the wave spectra such as significant wave height, mean period or direction were calculated and stored at every full model hour while wave spectra were only kept every three hours. Boundary conditions were transferred from the coarse to the fine grid every hour using the full modelled wave spectra. For both grids, the model was set-up and integrated in shallow water mode including depth refraction and depth-induced wave breaking. In the coarse grid simulation also monthly sea ice conditions from the northeast North Atlantic were included to account for the varying fetch arising from variations in sea ice coverage.

## 3 External data

### 3.1 Forcing data

For both, the coarse and the fine grid wind-wave simulations near-surface marine wind fields at 10m height were used. These were obtained from the high-resolution regional atmospheric hindcast described in (Geyer, 2014) as part of coastDat2. For the production of this atmospheric hindcast, the regional atmosphere model COSMO-CLM (Rockel et al. 2005) with a spectral nudging scheme (von Storch et al., 2000) was used to dynamically downscale the global atmospheric conditions given by the driving NCEP/NCAR reanalysis (Kalnay et al., 1996; Kistler et al., 2001). The downscaling was performed for the North Atlantic/ European region and the model was integrated on a rotated grid with a grid size of 0.22°x 0.22°corresponding to a spatial resolution of about 25-by-25 km. Atmospheric wind fields to drive the wave model were available and used every model hour. Validation of the atmospheric hindcast is described in Geyer (2014). Compared to the driving global reanalysis an improved representation of marine wind speeds in coastal areas, especially for higher wind speeds, is noted (Geyer et al., 2015).

For the coarse grid simulation, in addition, monthly sea ice concentrations provided by the Hadley Centre Sea Ice and Sea Surface Temperature data set (HadISST1.1, Rayner et al., 2003) were used. The sea ice concentrations had a spatial resolution of 1°x 1°and were spatially interpolated to the coarse wave model grid. Subsequently in the simulation sea ice was accounted for by treating all model grid points with sea ice concentrations exceeding 50 % as land for the corresponding time steps.

### 3.2 Reference data

A number of in situ wind-wave observations from platforms and buoys in the North Sea originating from different sources were available for validation (Table 1). Basic quality control was applied but no attempt to check homogeneity was done. The data cover different time periods including gaps and differ in their temporal resolution. Because the model output is available only at full model hours, the comparison with observations was limited to data that were measured within ±10 minutes around full hours.



In addition, wind-wave data derived from satellite provided by the merged altimeter wave height database version 11.0 (Queffeulou, 2013) were used. These data originate from the GlobWave project (http://www.globwave.org) and cover the period 1991 to present. The satellite data were co-located with the model data using a co-location criteria of $\pm10$ minutes at which the position of the satellite was matched with the nearest grid point of the wave model.

To compare the wave model results in space and time also data from the ERA-Interim global reanalysis (Dee et al., 2011) spatially interpolated to the coastDat2 wave model grid were used. More specifically, data from the ocean wave product of ERA-Interim were used, which are available every six hours at a spatial resolution of $0.75°$ latitude $\times$ $0.75°$ longitude. Data are available from 1980 onwards. From 1990 onwards wave spectra in ERA-Interim were adjusted using altimeter data (Dee et al., 2011).

In order to avoid biases, only those instances in time are used for comparison for which both, observations and hindcast data were simultaneously available. In particular, this results in comparison of hourly/six-hourly data when coastDat2 hindcast/ERA-Interim reanalysis is involved.

## 4    Evaluation

For the evaluation of the wave hindcast the following error metrics were applied: the mean, the standard deviation (SD), the 15 bias, the root mean square error (RMSE), the scatter index (SI), the correlation coefficient (r). To evaluate the unbiased RMSE the standard deviation of the error (SDE) was used. All measures were calculated for the entire datasets if not stated otherwise. In the following, measures are called errors when simulations are compared with observations and are called differences when the two simulations are compared. Further details of the error metrics can be found in Appendix A.

### 4.1    Significant wave height

Significant wave height (SWH) derived from hindcast data and in situ measurements was compared at seven sites in the North Sea (Figure 2). Comparison of instantaneous values revealed noticeable scatter between modelled and observed data with biases ranging between about 0 and 0.25 m. Root mean square errors varied between approximately 0.4 and 0.7 m (Table 2). When distributions were compared, generally a reasonable agreement was inferred for the lower to intermediate percentiles corresponding to wave heights of up to $2-4$ m depending on location (Figure 2). Higher percentiles were generally found to 25 be overestimated in the hindcast data.

    To put these findings into perspective Figure 2, in addition, shows a corresponding analysis for SWH derived from the ERA-Interim reanalysis. Here generally a tendency towards an underestimation of the higher SWHs can be inferred. This feature is most pronounced at near coastal locations such as at station ELB. An evaluation with a normalised Taylor-diagram (Figure 3) indicates that for both, the ERA-Interim and the coastDat2 SWHs correlation with observations typically vary around 30 about 0.9 with the values being slightly higher for the ERA-Interim reanalysis. The analysis further reveals that the observed SWH variability is somewhat underestimated by the ERA-Interim reanalysis and overestimated in the coastDat2 data set. The centred root mean square errors are slightly above (below) 0.5 m for coastDat2 (ERA-Interim). No substantial differences in





conclusions are obtained when 6-hourly instead of hourly values for the coastDat2 SWHs are used. A more detailed comparison of error statistics for the seven locations is provided in Table 2.

A spatial comparison of the differences between ERA-Interim and coastDat2 SWHs for the common period 1980 – 2014 is illustrated in Figure 4. With the exception of near-coastal waters, the SWH obtained from the coastDat2 hindcast is on average

higher compared to that from the ERA-Interim reanalysis with increasing differences from south to north. Largest systematic differences of up to 0.6 m were found off the Norwegian coast. A similar spatial pattern is obtained when root mean square differences are compared. These differences were found to vary between about 0.4 m in the southern North Sea and more than 0.9 m in the northern parts of the model domain. When the bias is removed from the root mean square differences, again a similar pattern for the standard deviation of differences with somewhat smaller values compared to the root mean square

differences is obtained.

When the spatial comparison is made with the SWHs derived from the GlobWave satellite dataset instead of ERA-Interim, differences are less distinct but still present (Figure 5). Similar spatial features but with smaller values are found in the southern North Sea and near the coasts. The latter corresponds to too small wave heights in ERA-Interim when compared to GlobWave data (not shown). Although the underestimation may reach values of up to 0.4 m on average, the magnitude of the systematic

differences between ERA-Interim and GlobWave is still smaller than that between the coastDat2 and the GlobWave data. Note that the robustness of these results is limited, as the number of available satellite data is small. For most of the domain less than 100 co-located data points were available for the comparison. This corresponds to only about one-tenth of a percent of the potentially available hourly values within the period 1992 – 2014.

To obtain a more robust figure an additional comparison of co-located data from GlobWave and coastDat2 was made taking

all co-located data irrespective of their location into account. The results are shown in Figure 6 and corresponding error statistics are presented in Table 2. Altogether, errors statistics obtained and conclusions derived from this exercise are similar to those based on comparison with in situ observations. Values for bias and RMSE are slightly enhanced compared to those derived from comparison with in situ observations. This can be attributed to the fact that most of the in situ observations were taken in the southern part of the North Sea where error statistics of the coastDat2 hindcast are smaller, while the more northern parts

with larger errors have a stronger weight in the GlobWave comparison.

## 4.2 Wave period

Different definitions for wave periods are used depending on the specific analysis. Here two of the more frequently used measures are used for comparison with observations: (i) the mean zero crossing period derived from the zeroth and second-order moment of the spectrum ($T_{m02}$) and (ii) the mean wave period (MWP) defined by the ratio of the first and zeroth-order

moment of the spectrum corresponding to the total energy of the wave spectrum (Holthuijsen, 2007). The Comparison is made based on the availability of the different data sets.

When observed and hindcast $T_{m02}$ periods are compared, an underestimation (overestimation) for short (long) periods can be inferred for most locations (Figure 7). Deviations are mostly small and typically in the order of 0.5 s, except for the near shore location WES. Despite the scatter for instantaneous values, hindcast wave periods on average show a good agreement





with the observations. The larger deviations occurring at the location WES are probably related to the relatively shallow water depth which might lead to too small wave dissipation caused by missing small scale bathymetric features not resolved at the given spatial resolution of the wave hindcast model.

A spatial comparison of the MWPs from the coastDat2 hindcast and the ERA-Interim reanalysis for the period 1980–2014 is presented in Figure 8. On average the coastDat2 hindcast shows longer MWP (up to more than 0.6 s) for large areas of the North Sea with the largest differences occurring in the deeper waters in the northeastern part of the model domain. In coastal areas, differences are mostly less pronounced. For the root mean square differences and the standard deviation of the differences similar spatial patterns can be inferred.

A comparison between $T_{m02}$ period measures derived from observations and the coastDat2 hindcast is presented in Table 3. The MWP measures between observations, coastDat2 hindcast and the ERA-Interim reanalysis is presented in Table 4. It can be inferred, that compared to observations $T_{m02}$ periods are on average underestimated in the coastDat2 hindcast while MWPs are on average overestimated. For ERA-Interim no $T_{m02}$ data were available. MWPs are similarly biased high although with somewhat smaller values. Variability is generally overestimated in both model simulations with the ERA-Interim errors again being smaller.

## 4.3 Wave direction

A comparison between mean wave directions derived from coastDat2, ERA-Interim and observations was performed at three locations (Figure 9). Generally, a good agreement was inferred. At FN1 a more systematic deviation for mean wave directions coming from southerly directions was obtained. Both model data, coastDat2 and ERA-Interim, show the same systematic error. This may point to some bathymetric effects that remain unresolved at the chosen model resolution or to the installation of wave measurements that might be sheltered from certain wave directions.

## 4.4 Individual extreme events

In order to illustrate the amount of agreement and disagreement described above in a more direct and accessible way, modelled and observed wave parameters during an extreme wave event were compared. Here we used the event generated by the extra-tropical storm Britta (31 October 2006 – 1 November 2006) which caused some structural damage at the platform FINO1 (Kettle, 2015). Visual inspection of time series of significant wave height, wave periods and direction reveals that while the overall development for the days around the storm is reasonably captured by both, the coastDat2 hindcast and the ERA-Interim reanalysis, both data sets substantially underestimate the peak significant wave height that occurred during the storm (Figure 10).

The underestimation is less severe in the coastDat2 hindcast and substantially more pronounced in the ERA-Interim reanalysis. Inspection of the time series further shows, that during less stormy periods ERA-Interim data are generally closer to the observations while positive systematic errors are obvious for the coastDat2 hindcast. In detail and for the time series shown it was found that (i) peak significant wave heights appear to be better represented in the coastDat2 hindcast while there generally appears to be a too rapid increase towards the extremes, (ii) during times with small wave periods the $T_{m02}$ period appears to



be reasonable in the coastDat2 simulation while again a too rapid increase towards the extremes is observed, (iii) for the MWP a systematic bias towards too high values is clearly visible that is substantially less pronounced in the ERA-Interim data and (iv) mean wave directions are very well represented in both simulations.

Extreme value analyses are often based on maximum values occurring within an interval for which a given threshold is exceeded. For the analysis, the exact timing of the extreme within this interval is less important. In order to assess the representativeness of such extremes in the modelled data set, we defined the duration of an extreme event as the time period for which the SWH exceeded the 95th percentile of the observed SWH and sampled the maximum SWHs that occurred during that interval from all data sets; that is, the observations, coastDat2 and ERA-Interim. The analysis was performed exemplary for station FN1. It was found that coastDat2 overestimated the observed maximum SWH by 0.38 m with an RMSE of 1.12 m. ERA-Interim on the other hand underestimated the observed maximum SWH by -0.63 m but showed a smaller RMSE of 0.81 m. While the underestimation of ERA-Interim extremes could partly be related to the coarser temporal resolution of six hours, the result indicates that for extreme value analyses data from coastDat2 would provide a more conservative estimate.

## 5  Conclusions

To our knowledge, the described data set represents the longest regional reconstruction of wind-wave climate for the North Sea based on dynamically downscaled high-resolution atmospheric reanalysis data. It covers more than 60 years and provides hourly integrated wind-wave parameters and spectral wave information every three hours at a spatial resolution of approximately 3-by-3 nautical miles.

Altogether the reconstructed data show a good agreement with observations although the shorter and coarser ERA-Interim reanalysis outperforms the hindcast in many aspects. The main advantages of this coastDat2 wind-wave hindcast are the extended period for which reconstructed data are available, the increased resolution both in space and time and the more comprehensive set of wave parameters available (Appendix B). In addition, the hindcast provides a more conservative estimates for extreme value analyses which in some circumstances might be of advantage. As an example, one might think of safety related assessments in offshore engineering. As part of a comprehensive approach to consistently reconstruct atmospheric (Geyer, 2014), tide-surge (Gaslikova and Weisse, 2013) and wind-waves conditions (this paper), the hindcast contributes to a consistent met-ocean data set that provides a useful data source for analysing long-term meteo-marine climate variability and change. Furthermore, it represents a valuable source of data for a large variety of related and more applied research in the industry and administration (Weisse et al., 2015). As part of the coastDat2 dataset, data from the described wind-wave hindcast (Groll and Weisse, 2016) are freely available and can be accessed from doi:10.1594/WDCCcoastDat2_WAM–North_Sea.





## Appendix A:  Error metrics

To evaluating the skill of the wave model standard statistical measures are used. The mean, the standard deviation ($SD$), the bias, the root mean square error ($RMSE$), the scatter index ($SI$) and the correlation coefficient ($r$) are defined as follows

$$mean = \frac{\sum_{i=1}^{n} E_i}{n} \tag{A1}$$

$$SD = \sqrt{\frac{\sum_{i=1}^{n}(E_i - \overline{E})^2}{n}} \tag{A2}$$

$$bias = \frac{\sum_{i=1}^{n}(E_i - R_i)}{n} \tag{A3}$$

$$RMSE = \sqrt{\frac{\sum_{i=1}^{n}(E_i - R_i)^2}{n}} \tag{A4}$$

$$SI = \frac{RMSE}{\overline{R}} \tag{A5}$$

$$r = \frac{\sum_{i=1}^{n}(E_i - \overline{E})(R_i - \overline{R})}{\sqrt{\sum_{i=1}^{n}(E_i - \overline{E})^2 \sum_{i=1}^{n}(R_i - \overline{R})^2}} \tag{A6}$$

where the overlines indicate mean values over time, $n$ denotes the number of data pairs, $E$ refers to the estimator (coastDat2 or ERA-Interim) and $R$ refers to the reference data (in situ or satellite observations). If coastDat2 is compared to ERA-Interim, the later is used as the reference data $R$. Using the decomposition of the mean squared error ($MSE$) into variance ($VAR$) and bias ($MSE = VAR + bias^2$), the standard deviation of the error ($SDE$) is used as an estimate for the unbiased $RMSE$ and is defined as

$$SDE = \sqrt{RMSE^2 - ME^2} \tag{A7}$$

## Appendix B:  Available variables

See Table B1





*Acknowledgements.* We thank the providers of the observational data, namely, the Bundesamt für Seeschifffahrt und Hydrographie (BSH), the Norwegian Meteorological Institute, the Royal Netherlands Meteorological Institute (KNMI), the European Space Agency (ESA) and the French Research Institute for Exploitation of the Sea (IFREMER) for their support. We also thank the providers of the numerical data, namely, the European Centre for Medium-Range Weather Forecasts (ECMWF) for the ERA-Interim reanalysis data and the Met Office

5    Hadley Centre for Climate Science and Services for access to the sea ice data set.



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



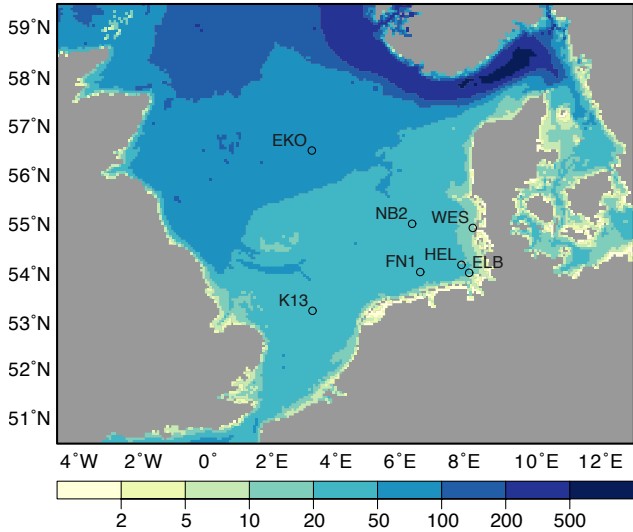

**Figure 1.** Model domain and bathymetry used for the fine grid simulation. Colours indicate water depth in meters. Circles mark observational sites used for the model evaluation. For the abbreviations see Tab. 1.

**Table 1.** Location and description of in situ measurements used for comparison.

| site | location | depth [m] | time period | type |
|------|----------|-----------|-------------|------|
| EKO | 2.6°E / 56.5°N | 72 | 1980-1998 | platform |
| K13 | 3.2°E / 53.2°N | 25 | 1980-2008 | platform |
| NB2 | 6.3°E / 55.0°N | 42 | 1993-2012 | buoy |
| FN1 | 6.6°E / 54.0°N | 30 | 2003-2012 | platform |
| HEL | 7.9°E / 54.2°N | 20 | 1989-2012 | buoy |
| ELB | 8.1°E / 54.0°N | 25 | 1990-2012 | buoy |
| WES | 8.2°E / 54.9°N | 14 | 1993-2012 | buoy |

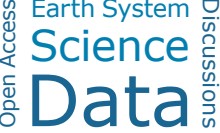

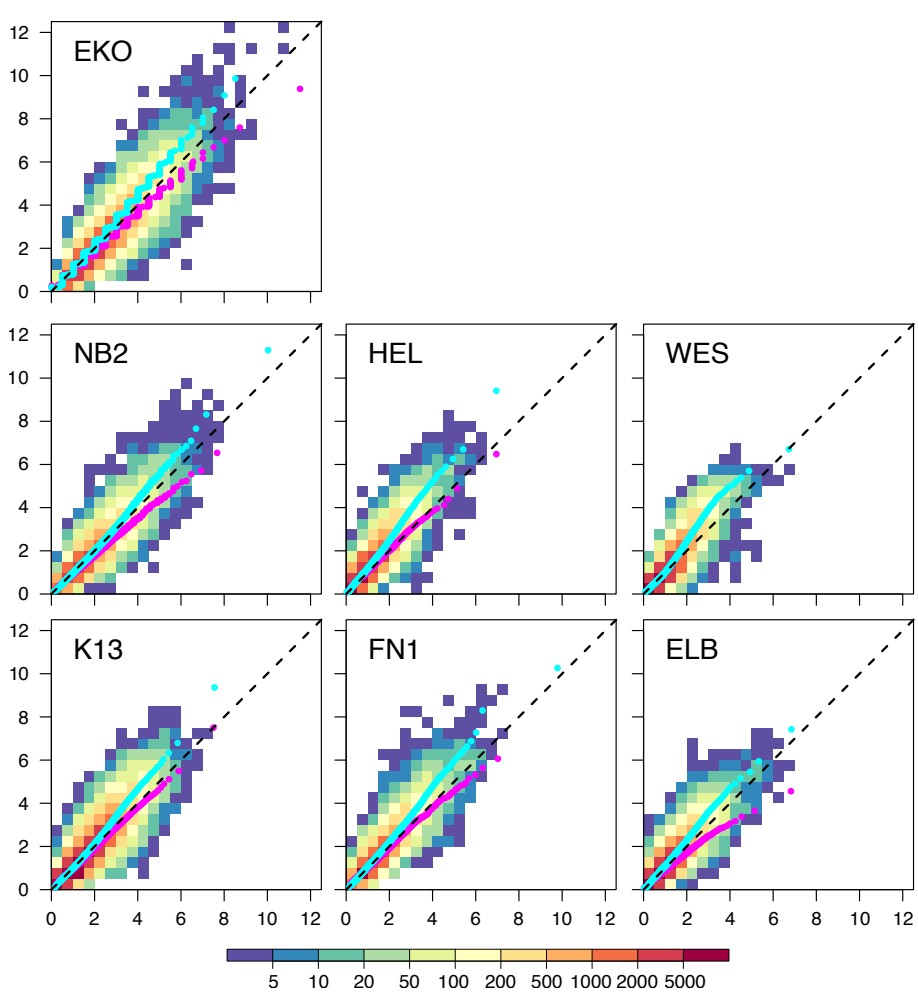

**Figure 2.** Scatter and quantile-quantile plot of observed (x-axis) and coastDat2 hindcast (y-axis) SWH [m]. Coloured squares indicate the number of data in each 0.5 m x 0.5 m bin. Coloured dots represent the percentile values of the quantile-quantile plots between observed and coastDat2 hindcast (cyan) and between observed and ERA-Interim reanalysis significant wave heights (magenta). Note that the comparison is made for different periods according to data availability (see Table 1).



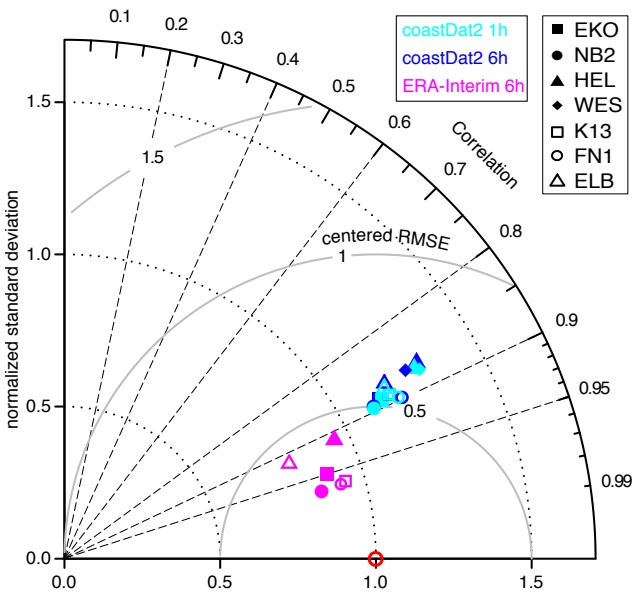

**Figure 3.** Normalized Taylor diagram from hourly (cyan) and six hourly (blue) coastDat2 SWHs and from six-hourly ERA-Interim (magenta) SWHs at seven (coastDat2) and at six (ERA-Interim) observational sites. The red circle would represent a perfect model compared to the observations.

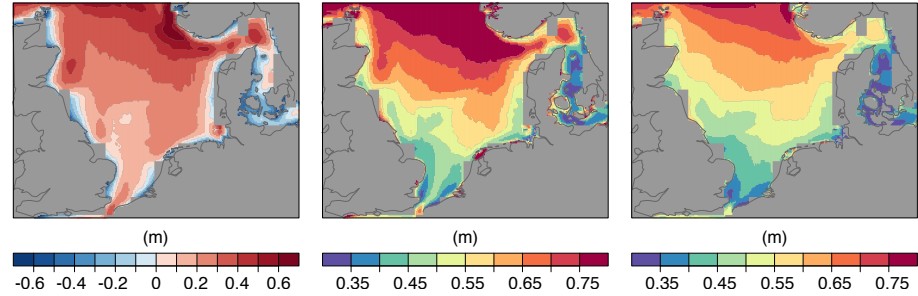

**Figure 4.** Spatial distribution of bias (left), root mean square distance (middle) and standard deviation of the error (right) [m] between SWHs derived from the coastDat2 hindcast and the ERA-Interim reanalysis for the period 1980–2014.





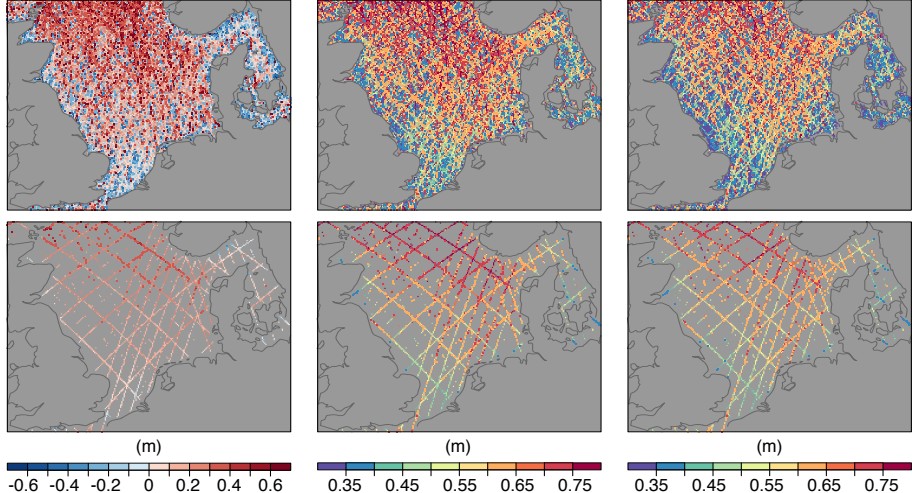

**Figure 5.** Spatial distribution of bias (left), root mean square error (middle) and standard deviation of the error (right) [m] between SWHs from the coastDat2 hindcast and the GlobWave data set for the period 1992 – 2014 when all available satellite data are used for comparison (top) and when only data with more than 100 flyovers per grid point are used (bottom).

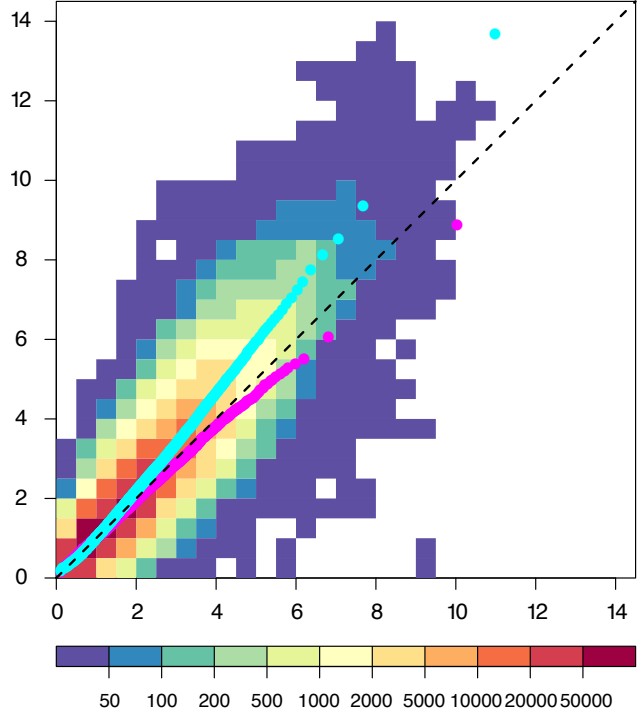

**Figure 6.** Scatter and quantile-quantile plot of observed (x-axis) and coastDat2 hindcast (y-axis) SWH [m]. Coloured squares indicate the number of the data in each 0.5 m x 0.5 m bin. Coloured dots represent the percentile values of the quantile-quantile plots between observed and coastDat2 hindcast significant wave heights (cyan) and between observed and ERA-Interim SWH (magenta).





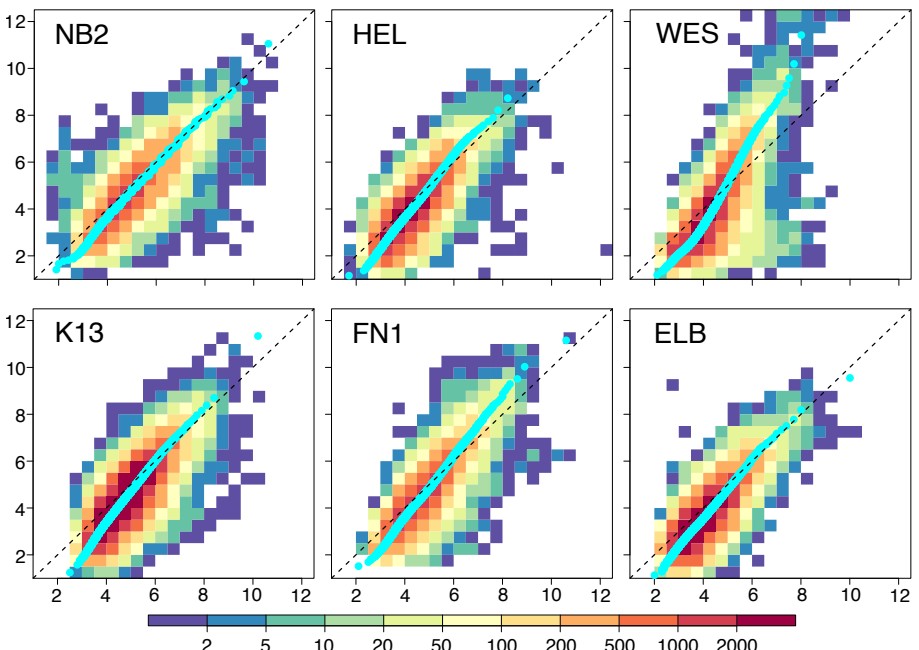

**Figure 7.** Scatter and quantile-quantile plot of observed (x-axis) and coastDat2 hindcast (y-axis) $T_{m02}$ wave period [s]. Coloured squares indicate the density of the data in each 0.5 s x 0.5 s bin. Coloured dots represent the percentile values of the quantile-quantile plots between observed and coastDat2 hindcast $T_{m02}$ (cyan) wave periods. Note that the comparison is made for different periods according to data availability (see Table 1).

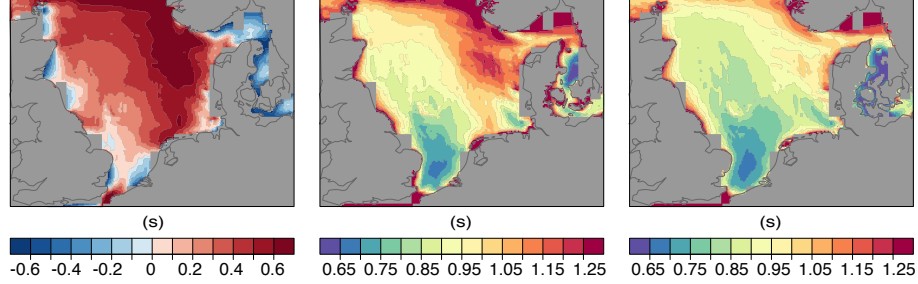

**Figure 8.** Spatial distribution of bias (left), root mean square error (middle) and standard deviation of the error (right) [s], between the mean wave period between from the coastDat2 hindcast and ERA-Interim reanalysis for the period 1980 – 2014.





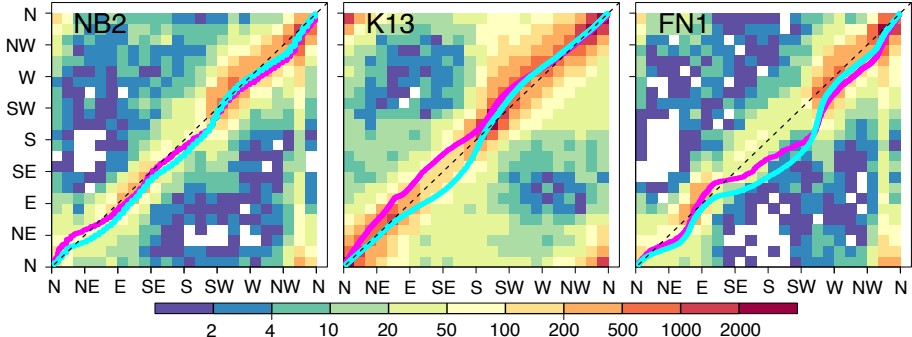

**Figure 9.** Scatter and quantile-quantile plot of observed (x-axis) and coastDat2 hindcast (y-axis) mean wave direction. Coloured squares indicate the number of counts in each 15°x 15° bin. Coloured dots represent the percentile values of the quantile-quantile plots between observed and coastDat2 hindcast mean wave directions (cyan) and between observed and ERA-Interim mean wave directions (magenta). Note that the comparison is made for different periods according to data availability (see Table 1).

**Table 2.** Error metrics for the SWH at seven locations derived from in situ observations (OBS), coastDat2 hindcast (CD2) and ERA-Interim reanalysis (ERAi), for hourly (coastDat2) and six-hourly (ERA-Interim) data respectively.

| name | count | mean [m] | standard deviation [m] | bias [m] | RMSE [m] | scatter index | correlation |
|------|-------|----------|------------------------|----------|----------|---------------|-------------|
| | CD2/ERAi | OBS/ CD2/ ERAi | OBS/ CD2/ ERAi | CD2/ ERAi | CD2/ ERAi | CD2/ ERAi | CD2/ ERAi |
| EKO | 51620/ 22589 | 2.08/ 2.24/ 1.94 | 1.29/ 1.48/ 1.15 | 0.16/ -0.14 | 0.71/ 0.44 | 0.34/ 0.21 | 0.88/ 0.95 |
| K13 | 84744/ 42372 | 1.5/ 1.59/ 1.43 | 0.92/ 1.08/ 0.86 | 0.09/ -0.07 | 0.51/ 0.26 | 0.34/ 0.17 | 0.88/ 0.96 |
| NB2 | 28485/ 5245 | 1.75/ 1.77/ 1.51 | 1.11/ 1.23/ 0.91 | 0.03/ -0.17 | 0.56/ 0.35 | 0.32/ 0.21 | 0.89/ 0.97 |
| FN1 | 30893/ 4655 | 1.52/ 1.68/ 1.5 | 0.97/ 1.16/ 0.9 | 0.16/ -0.03 | 0.56/ 0.26 | 0.37/ 0.17 | 0.89/ 0.96 |
| HEL | 46074/ 14208 | 1.09/ 1.34/ 1.15 | 0.76/ 0.98/ 0.71 | 0.25/0.06 | 0.55/ 0.32 | 0.5/ 0.29 | 0.87/ 0.91 |
| ELB | 55190/ 14292 | 1.04/ 1.18/ 0.92 | 0.73/ 0.85/ 0.56 | 0.13/ -0.12 | 0.44/ 0.32 | 0.42/ 0.31 | 0.87/ 0.92 |
| WES | 43979/ NA | 1.08/ 1.2/ NA | 0.72/ 0.94/ NA | 0.12/ NA | 0.48/ NA | 0.45/ NA | 0.88/ NA |
| SAT | 901451/ 82067 | 1.76/ 1.94/ 1.69 | 1.11/ 1.34/ 0.97 | 0.18/ -0.06 | 0.64/ 0.36 | 0.37/ 0.20 | 0.89/ 0.94 |

NA-not available



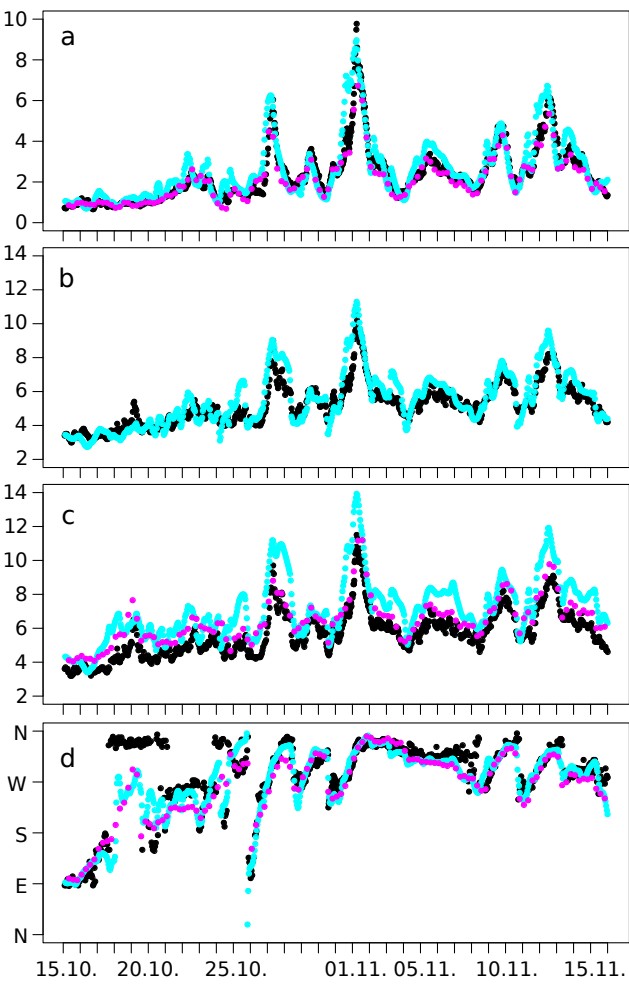

**Figure 10.** Time series of significant wave height [m] (a), $T_{m02}$ [s] (b) and mean wave period [s] (c) and mean wave direction [°] (d) at FN1 for the period 15 October 2006, 00:00 UTC to 15 November 2006, 00:00 UTC. Observations are shown in black (a–d), the coastDat2 hindcast is shown in cyan (a–d) and the ERA-Interim data are shown in magenta (a,c,d).



**Table 3.** Error metrics for the $T_{m02}$ period at six locations derived from in situ observations (OBS) and coastDat2 hindcast (CD2) data for hourly values.

| name | count | mean [s] | standard deviation [s] | bias [s] | RMSE [s] | scatter index | correlation |
|------|-------|----------|------------------------|----------|----------|---------------|-------------|
|      | CD2   | OBS/ CD2 | OBS/ CD2               | CD2      | CD2      | CD2           | CD2         |
| K13  | 84744 | 4.76 / 4.37 | 0.91/ 1.21          | -0.39    | 0.84     | 0.18          | 0.79        |
| NB2  | 28485 | 4.92/ 4.65  | 1.19/ 1.34          | -0.27    | 0.87     | 0.18          | 0.79        |
| FN1  | 30893 | 4.72/ 4.51  | 1.05/1.14           | -0.21    | 0.82     | 0.17          | 0.82        |
| HEL  | 46074 | 4.32/ 3.93  | 0.93/ 1.21          | -0.30    | 0.81     | 0.19          | 0.78        |
| ELB  | 55190 | 4.03/ 3.6   | 0.92/ 1.12          | -0.43    | 0.82     | 0.20          | 0.78        |
| WES  | 43979 | 4.22/ 3.83  | 0.89/ 1.48          | -0.40    | 1.07     | 0.25          | 0.76        |

**Table 4.** Error metrics for the MWP at four locations derived from in situ observations (OBS), coastDat2 hindcast (CD2) and ERA-Interim (ERAi) data for hourly (coastDat2) and six-hourly (ERA-Interim) data, respectively.

| name | count | mean [s] | standard deviation [s] | bias [s] | RMSE [s] | scatter index | correlation |
|------|-------|----------|------------------------|----------|----------|---------------|-------------|
|      | CD2/ERAi | OBS/ CD2/ ERAi | OBS/ CD2/ ERAi   | CD2/ ERAi | CD2/ ERAi | CD2/ ERAi    | CD2/ ERAi   |
| NB2  | 28485/ 5254 | 5.25/ 6.16/ 5.75 | 1.23/ 1.54/ 1.2 | 0.91/ 0.59 | 1.31/ 0.83 | 0.25/ 0.16 | 0.79/ 0.88 |
| FN1  | 30893/ 4655 | 5.11/ 6.14/5.75  | 1.19/ 1.66/ 1.21 | 1.04/ 0.62 | 1.41/ 0.8  | 0.28/0.16  | 0.82/ 0.91 |
| ELB  | 55190/ 14292 | 4.32/ 4.99/ 4.96 | 1.02/ 1.39/ 1.05 | 0.67/ 0.63 | 1.08/ 0.87 | 0.25/ 0.2 | 0.8/ 0.83 |
| WES  | 43979/ NA | 4.58/ 5.45/ NA | 1.02/ 1.77/ NA | 0.87/ NA | 1.46/ NA | 0.32/ NA | 0.78/ NA |

NA-not available



**Table B1.** List of available variables

|  | Variable name | Unit | Long name | Standard name |
|---|---|---|---|---|
| 1 | dd | degree | wind direction | wind_to_direction |
| 2 | ds | degree | total directional spread | sea_surface_wave_directional_spread |
| 3 | ds_sea | degree | sea directional spread | sea_surface_wind_wave_directional_spread |
| 4 | ds_swell | degree | swell directional spread | sea_surface_swell_wave_directional_spread |
| 5 | ff | $ms^{-1}$ | wind speed | wind_speed |
| 6 | fv | $ms^{-1}$ | friction velocity |  |
| 7 | hs | m | total significant wave height | sea_surface_wave_height |
| 8 | hs_sea | m | sea significant wave height | sea_surface_wind_wave_height |
| 9 | hs_swell | m | swell significant wave height | sea_surface_swell_wave_height |
| 10 | nws | none | normalised_wave_stress | normalised_wave_stress |
| 11 | thq | degree | total mean wave direction | sea_surface_wave_to_direction |
| 12 | thq_sea | degree | sea mean wave direction | sea_surface_wind_wave_to_direction |
| 13 | thq_swell | degree | swell mean wave direction | sea_surface_swell_wave_to_direction |
| 14 | tmp | s | total mean period | sea_surface_wave_mean_period_from_variance _spectral_density_inverse_frequency_moment |
| 15 | tmp_sea | s | sea mean period | sea_surface_wind_wave_mean_period_from_variance _spectral_density_inverse_frequency_moment |
| 16 | tmp_swell | s | swell mean period | sea_surface_swell_wave_mean_period_from_variance _spectral_density_inverse_frequency_moment |
| 17 | tm1 | s | total m1-period | sea_surface_wave_mean_period_from_variance _spectral_density_first_frequency_moment |
| 18 | tm1_sea | s | sea m1-period | sea_surface_wind_wave_mean_period_from_variance _spectral_density_first_frequency_moment |
| 19 | tm1_swell | s | swell m1-period | sea_surface_swell_wave_mean_period_from_variance _spectral_density_first_frequency_moment |
| 20 | tm2 | s | total m2-period | sea_surface_wave_mean_period_from_variance _spectral_density_second_frequency_moment |
| 21 | tm2_sea | s | sea m2-period | sea_surface_wind_wave_mean_period_from_variance _spectral_density_second_frequency_moment |
| 22 | tm2_swell | s | swell m2-period | sea_surface_swell_wave_mean_period_from_variance _spectral_density_second_frequency_moment |
| 23 | tp | s | total peak-period | sea_surface_wave_peak_period_from_variance_spectral_density |
| 24 | tp_sea | s | sea peak period | sea_surface_wind_wave_peak_period_from_variance_spectral_density |
| 25 | tp_swell | s | swell peak period | sea_surface_swell_wave_peak_period_from_variance_spectral_density |