# Peer review of "A multi-decadal wind-wave hindcast for the North Sea 1949 – 2014: coastDat2"

_Earth System Science Data, 2017_

## Referee Comment (RC1) · Anonymous Referee #1 · 30 Jul 2017

This manuscript presents a new multi-decadal wind-wave hindcast for the North Sea. The hindcast spans 1949-2014 and forms part of the CoastDat database. The hindcast provides a long record (60 years) at relatively high spatial resolution (3 nautical miles). The study assesses the skill of the hindcast relative to a set of in-situ platform and buoy wave observations in the North Sea. The earliest data to which a validation was carried out were two observation points (EKO and K13) both commencing in 1980. The performance of the hindcast was compared to the skill of wave fields derived from the ECMWF ERA-Interim reanalysis. Key differences between the datasets are displayed in Figure 6, where the CoastDat2 hindcast displays a tendency to overestimate observed wave heights, whereas ERA-Interim fields slightly underestimate significant wave heights. This is presented as a potential benefit for the database in providing

a conservative estimate for extreme wave heights in the North Sea for planning purposes. In general however, Table 2 suggests ERA-Interim has greater skill relative to observations than the presented CoastDat2 database.

Despite the ERA-Interim database demonstrating greater skill in the region of interest, the paper argues the extended time-series, the high spatial resolution, and the overestimated (conservative) extreme wave height estimates provide a distinctive dataset which will be of value to the user community in the North Sea. While the high spatial resolution offers value, I have reservations about the merit of the extended time-series given no investigation of the skill of the hindcast to represent trends (against long independent datasets) or of the homogeneity of the dataset has been carried out on the database. Given this shortcoming, my recommendation is the study undergo major revision before being published. This revision should include some consideration of the skill of the full temporal extent of the time-series.

Other than this consideration, the paper is generally well written and clear. I have only a few further minor comments which I would also like to see considered in the revised version of the manuscript.

Abstract: Second sentence. No statistics have yet been introduced. Some revision of English is required.

Figures 2, 6 and 7. Please label x and y axes, as opposed to including information in caption.

Figures 4, 5 and 8. Please provide some axis information, in order to resolve spatial scales.

An equivalent table to Table 2, 3 and 4 for directional information would add benefit.

---

## Referee Comment (RC2) · Anonymous Referee #2 · 16 Aug 2017

General Comments

This paper describes a surface wave hindcast dataset for the North Sea based on atmospheric reanalysis from 1949-2014. The primary motivation for the dataset is the discontinuation of the coastDat1 dataset in 2007. The write-up is generally clear, well organized, and fairly complete. The accessibility of the data online is convenient, though the file sizes are quite large (∼100 gb per file). This could be alleviated by splitting the files into yearly output, but I respect the judgment of the authors if their target audience is capable of downloading data of that size. I provide more specific comments on the write-up below and recommend to publish this work after minor revision.

Specific Comments

1. My primary concern relates to properly motivating the benefits of this new dataset in comparison to other options. coastDat2 is presented here as an alternative to the ERA-Interim reanalysis and the analysis of the model output is performed relative to the ERA-Interim data. In both cases, comparisons are made to several platform observations and satellite derived data products. The study finds that the ERA-Interim data performs better on average, with larger differences in performance under high wind events. The study argues that the coastDat2 dataset has merit relative to the ERA-Interim data because (i) it provides 30 additional years of hindcast for understanding climate trends, (ii) it is a higher resolution dataset, (iii) it includes additional output parameters, and (iv) it gives a more conservative dataset in the sense that wave heights are generally overestimated.

The first point is clear, but would benefit from comparison with published coastDat1 to show if the data trends are consistent/reliable. The second point intuitively makes sense, but is not supported by the results. The higher resolution data does not seem to provide any obvious benefit at the observation locations (e.g. mixed results in Figure 2). If the authors wish to argue this point they should offer evidence that there is useful information in the high-resolution output (For example, this is contradicted by the statement at the end of page 4 that "No substantial differences in conclusions are obtained when 6-hourly instead of hourly values for the coastDat2 SWHs are used."). The third point may be true, but little description is offered of the additional data products. I suggest expanding appendix B to comment on the additional fields including their meaning and why they are useful. I do not follow the logic behind the fourth point. If anything, the consistent positive bias in coastDat2 suggests a problem in either model physics or the forcing data. Was any attempt made to understand the source of this bias and/or correct for it? Has this version of WAM (with the same wind-input and dissipation source terms) been validated in these conditions in other locations to show it is appropriate for simulating large wave heights? Are there similar biases in the wind field? These points should be commented on rather than trying to argue that this bias may be a good thing for some applications.

2. Please comment more on ERA-Interim data (Page 4 Line 5). Is the same wave model used? What data does it assimilate (is it the same data used in the verification, are errors consistent in the pre and post assimilation years)? What resolution was the ERA-Interim atmospheric forcing? Are the 6 hour outputs instantaneous or 6 hour mean?

3. The wave results conclude in 2014, while the coastDat2 atmospheric results conclude in 2012. It should be explicitly mentioned if the time series was extended the additional years in a consistent manner.

Technical corrections

P1L16: ‘Data from wind-wave hindcast data’ <- Remove 2nd “data”

P2L11: ‘based on’ <- What does “based on” mean here?

P2L13: ‘Weisse and Günther (2007)’ <- use parenthetical citation

P2L21: ‘following-up’ <- ‘follow-up’

P3L6: ‘only’ <- Remove

P3L8: ‘also’ <- Remove

P3L10: Are these neutral winds? Is atmospheric stability accounted for?

P3L13: ‘(Geyer, 2014)’ <- Use in-line citation

P3L22: ‘, in addition,’ <- Remove

P4L5: Remove ‘also’

P4L8: ‘onwards’ <- onwards to what? 2017?

Appendix A: The presentation here could be improved. E.g. in equation A1 isn’t mean=‘\overbar{E}’? It may be clearer to replace this.

What is ME in equation A7?

Appendix B: Some details about what the fields actually represent could be useful. E.g., How is significant wave height computed (and other parameters)? How is significant wave height (and other parameters) partitioned into wind-sea and swell components?

Figure 1: Add a figure title, e.g. 'North Sea Bathymetry' and include units label with colorbar.

Figure 2: Add axes labels. Indicate in caption that these are the platform observations.

Figure 4: Please include x/y axes.

Figure 5: Please include x/y axes.

Figure 6: Add axes labels. Indicate in caption that these are the satellite observations.

Figure 7: Add axes labels. Indicate in caption that these are the platform observations.

Figure 8: Please include x/y axes.

Figure 9: Add axes labels. Indicate in caption that these are the platform observations.

Figure 10: Add axes labels. A legend in 1 panel would make the figure easier to understand than describing in the captions.

---

## Author Comment (AC1) · 29 Sep 2017

Reply to the comments of anonymous **reviewer #1** on the manuscript *essd-2017-36*
**"A multi-decadal wind-wave hindcast for the North Sea 1949–2014: coastDat2"** by Nikolaus Groll and Ralf Weisse

We thank the anonymous reviewer #1 for her/his comments, which helped to improve the manuscript.

In the following, the referee's comments are shown in blue.

*This manuscript presents a new multi-decadal wind-wave hindcast for the North Sea. The hindcast spans 1949-2014 and forms part of the CoastDat database. The hindcast provides a long record (60 years) at relatively high spatial resolution (3 nautical miles). The study assesses the skill of the hindcast relative to a set of in-situ platform and buoy wave observations in the North Sea. The earliest data to which a validation was carried out were two observation points (EKO and K13) both commencing in 1980. The performance of the hindcast was compared to the skill of wave fields derived from the ECMWF ERA-Interim reanalysis. Key differences between the datasets are displayed in Figure 6, where the CoastDat2 hindcast displays a tendency to overestimate observed wave heights, whereas ERA-Interim fields slightly underestimate significant wave heights. This is presented as a potential benefit for the database in providing a conservative estimate for extreme wave heights in the North Sea for planning purposes. In general however, Table 2 suggests ERA-Interim has greater skill relative to observations than the presented CoastDat2 database.*
*Despite the ERA-Interim database demonstrating greater skill in the region of interest, the paper argues the extended time-series, the high spatial resolution, and the overestimated (conservative) extreme wave height estimates provide a distinctive dataset which will be of value to the user community in the North Sea. While the high spatial resolution offers value, I have reservations about the merit of the extended time-series given no investigation of the skill of the hindcast to represent trends (against long independent datasets) or of the homogeneity of the dataset has been carried out on the database. Given this shortcoming, my recommendation is the study undergo major revision before being published. This revision should include some consideration of the skill of the full temporal extent of the time-series.*

The main concern of the reviewer is "reservations about the merit of the extended time-series given no investigation of the skill of the hindcast to represent trends (against long independent datasets) or of the homogeneity of the dataset has been carried out on the database". The argument is reasonable although, because of the lack of long-term wave measurements, it can be addressed only indirectly: The wave climate strongly depends on the wind climate and the quality and homogeneity of the wave hindcast is thus strongly coupled to those of the driving atmospheric data which are discussed in Geyer (2014) and in Geyer et al. (2015). In these papers, hindcast wind is compared with observations and for most locations, a slight overestimation and root mean square errors between 1.6 ms$^{-1}$ and 3.4 ms$^{-1}$ were found. Moreover, the long-term (decadal) variability described closely resembles that known from variations in storm activity over the North Sea. The latter was shown to occur also for waves in a hindcast that uses wind fields from a different regional atmosphere model that was, however, driven by the same global reanalysis (Weisse and Günther (2006).

Regarding homogeneity Kistler et al. (2001) investigated the issue for the global NCEP reanalysis that provided the boundaries for the regional atmosphere models used in this and the study of Weisse and Günther (2006). Kistler et al. (2001) showed that for the Northern Hemisphere rapidly increased from 1948 to 1958 while there is only a small increase afterwards. The latter implies that the driving global reanalysis is mostly homogeneous for our area after about 1958 and as there were no changes in regional model systems used in Geyer (2014) and our study we conclude that the same also holds for our wave hindcast.

To address the reviewers concern we have added a corresponding paragraph in the revised manuscript.

We have not included analysis of long-term variability and trends as the main focus of this manuscript is to introduce this new wave hindcast. For the brevity of the manuscript, we focused on the overall capability of the wave hindcast to represent the wave climate and statistics. Value to users of such long-term data sets is described in Weisse et al. (2009) and Weisse et al. (2015).

*Other than this consideration, the paper is generally well written and clear. I have only a few further minor comments which I would also like to see considered in the revised version of the manuscript.*
*Abstract: Second sentence. No statistics have yet been introduced.*

We modified the corresponding sentence in the abstract accordingly.

*Some revision of English is required.*

We carefully checked the manuscript for possible errors.

*Figures 2, 6 and 7. Please label x and y axes, as opposed to including information in caption.*

We added axis labels to the figures.

*Figures 4, 5 and 8. Please provide some axis information, in order to resolve spatial scales.*

We added an axis scale to the figures.

*An equivalent table to Table 2, 3 and 4 for directional information would add benefit.*

We added a table with the error metrics for the wave direction.

References:

[revised manuscript text omitted]

For the coarse grid simulation  monthly sea ice concentrations provided by the Hadley Centre Sea Ice and Sea Surface Temperature  dataset (HadISST1.1, Rayner et al., 2003) were used. The sea ice concentrations had a spatial resolution of 1° x 1°and were spatially interpolated to the coarse wave model grid. Subsequently in the simulation sea ice was accounted for by treating all model grid points with sea ice concentrations exceeding 50 % as land for the corresponding time steps.

**3.2   Reference data**

A number of in situ wind-wave observations from platforms and buoys in the North Sea originating from different sources were available for validation (Table 1). Basic quality control was applied but no attempt to check homogeneity was done. The

data cover different time periods including gaps and differ in their temporal resolution. Because the model output is available only at full model hours, the comparison with observations was limited to data that were measured within ±10 minutes around full hours.

In addition, wind-wave data derived from satellite provided by the merged altimeter wave height database version 11.0 (Queffeulou, 2013) were used. These data originate from the GlobWave project (http://www.globwave.org) and cover the period 1991 to present. The satellite data were co-located with the model data using a co-location criteria of ±10 minutes at which the position of the satellite was matched with the nearest grid point of the wave model.

To compare the wave model results in space and time  data from the ERA-Interim global reanalysis (Dee et al., 2011) spatially interpolated to the coastDat2 wave model grid were used. More specifically, data from the ocean wave product of ERA-Interim were used, which employs the same wave model as used in this study. ERA-Interim wave data are available every six hours at 00, 06, 12 and 18 UTC and at a spatial resolution of 0.75° latitude × 0.75° longitude.  Wind data from satellites and in situ observations are assimilated into ERA-Interim. Further, from 1990 onwards wave spectra  were adjusted using altimeter data Wave buoy data were not used in the assimilation procedure (Dee et al., 2011). For comparison, we use the ERA-Interim wave data from 1980 to 2014.

[revised manuscript text omitted]

 The shorter and coarser ERA-Interim reanalysis  shows smaller errors for most locations and outperforms the coastDat2 hindcast in open waters. For coastal locations, for which unresolved bathymetry and land-sea distribution in ERA-Interim may play a role, improvements in coastDat2 are visible. Whereas the ERA-Interim wave product generally underestimates extreme significant wave heights, the coastDat2 hindcast has a tendency to overestimate. This is consistent with results from an analysis of marine wind fields used as forcing data which also showed some overestimation of the more severe wind speeds (Geyer et al., 2015). Compared to its predecessor the coastDat1 wave hindcast Weisse and Günther (2007), which terminates in 2007, the coastDat2 hindcast shows similar and comparable performance (not shown).

One benefit of the coastDat2 wind-wave hindcast  is the extended period for which reconstructed wave data are available. In particular, the use for extreme value statistics or assessment of long-term variability and change require data sets as long as possible. Unfortunately, wave measurements to validate the hindcast over such extended periods of time are unavailable. There are, however, some indications that the data may be used for such

5 analyses: The wave climate strongly depends on the wind climate and the quality and homogeneity of the wave hindcast is thus strongly coupled to those of the driving atmospheric data which are discussed in Geyer (2014) and in Geyer et al. (2015). In these papers, hindcast wind is compared with observations and for most locations, a slight overestimation and root mean square errors between $1.6\,\mathrm{ms}^{-1}$ and

10  $3.4\,\mathrm{ms}^{-1}$ were found. Moreover, the described long-term (decadal) variability, closely resembles the known variations in storm activity over the North Sea. The latter was shown to occur also for waves in a hindcast that uses wind fields from a different regional atmosphere model that was, however, driven by the same global reanalysis (Weisse and Günther, 2007).

Regarding homogeneity Kistler et al. (2001) investigated the issue for the global NCEP reanalysis that provided the boundaries

15 for the regional atmosphere models used in this and the study of Weisse and Günther (2007). Kistler et al. (2001) showed that the skill for the Northern Hemisphere rapidly increased from 1948 to 1958 while there is only a small increase afterwards. The latter implies that the driving global reanalysis is mostly homogeneous for our area after about 1958 and, as there were no changes in regional model systems used in Geyer (2014) and our study, we conclude that the same may also hold for our wave hindcast.

[revised manuscript text omitted]

---

## Author Comment (AC2) · 29 Sep 2017

Reply to the comments of anonymous **reviewer #2** on the manuscript *essd-2017-36*

**"A multi-decadal wind-wave hindcast for the North Sea 1949–2014: coastDat2" by Nikolaus Groll and Ralf Weisse**

We thank anonymous reviewer #2 for her/his comments, which helped to improve the manuscript.

In the following, the referee's comments are shown in blue.

*General Comments*
*This paper describes a surface wave hindcast dataset for the North Sea based on atmospheric reanalysis from 1949-2014. The primary motivation for the dataset is the discontinuation of the coastDat1 dataset in 2007. The write-up is generally clear, well organized, and fairly complete. The accessibility of the data online is convenient, though the file sizes are quite large (~100 gb per file). This could be alleviated by split- ting the files into yearly output, but I respect the judgment of the authors if their target audience is capable of downloading data of that size. I provide more specific comments on the write-up below and recommend to publish this work after minor revision.*

The World Data Centre for Climate (WDCC) as the distribution point of the presented hindcast provides via its web interface an easy way to generate yearly subsets of the dataset once one has registered. Further, the WDCC supports an API interface to generate automatic subsets via a shell script and ftp(sftp). The organization of data in files was based on previous experiences gained from the distribution of coastDat1.

*Specific Comments*
*My primary concern relates to properly motivating the benefits of this new dataset in comparison to other options. coastDat2 is presented here as an alternative to the ERA-Interim reanalysis and the analysis of the model output is performed relative to the ERA-Interim data. In both cases, comparisons are made to several platform observations and satellite derived data products. The study finds that the ERA-Interim data performs better on average, with larger differences in performance under high wind events. The study argues that the coastDat2 dataset has merit relative to the ERA- Interim data because (i) it provides 30 additional years of hindcast for understanding climate trends, (ii) it is a higher resolution dataset, (iii) it includes additional output parameters, and (iv) it gives a more conservative dataset in the sense that wave heights are generally overestimated.*

*The first point is clear, but would benefit from comparison with published coastDat1 to show if the data trends are consistent/reliable.*

We agree that a comparison with the previous hindcast coastDat1 (Weisse and Günther 2007) has benefits for the interpretation of the new hindcast coastDat2, but for brevity, we decided to not include another reference dataset. However, to address the point raised by the reviewer we included some comparison at the end of this reply. It shows that both simulations are rather similar in terms of error statistics. Computing annual Brier skill scores (defined as unity minus the ratio of error variances of both hindcasts) shows that these scores fluctuate around zero indicating comparable quality and performance of both wave hindcasts (Fig. 1R). Note that the comparison is limited to the southern North Sea. We added a corresponding statement to the revised manuscript.

*The second point intuitively makes sense, but is not supported by the results. The higher resolution data does not seem to provide any obvious benefit at the observation locations (e.g. mixed results in Figure 2). If the authors wish to argue this point they should offer evidence that there is useful information in the high-resolution output (For example, this is contradicted by the statement at the end of page 4 that "No substantial differences in conclusions are obtained when 6-hourly instead of hourly values for the coastDat2 SWHs are used.").*

We do see the point raised by the reviewer which was not well elaborated in the manuscript. In particular, for mean conditions and open sea areas, higher temporal and spatial resolution does not necessarily lead to an improvement when compared to ERA-Interim. For coastal locations for which bathymetry and land-sea distribution unresolved in ERA-Interim play a role improvements are visible. For example, for the station ELB, coastDat2 clearly better represents the observed distribution, in particular for high wave heights (Figure 2). Stations like WES are land points in ERA-Interim and are hardly represented by the nearest wet grid point. Potential benefits of the increased resolution in time are shown in Figure 10. Due to the coarser temporal resolution, ERA-Interim missed the peak in this case and underestimated the maximum by about 3m.

We have rewritten the manuscript to make these points more clear and explicit.

*The third point may be true, but little description is offered of the additional data products. I suggest expanding appendix B to comment on the additional fields including their meaning and why they are useful.*

We agree but found to the replication of such information tedious and beyond the scope of the manuscript. Detailed information and description of the wave parameters are available from can the WAM documentation referred to in the manuscript, or at  https://github.com/mywave/WAM/blob/master/documentation/MyWave_D1.1.pdf.
We added a corresponding sentence to the manuscript.

*I do not follow the logic behind the fourth point. If anything, the consistent positive bias in coastDat2 suggests a problem in either model physics or the forcing data. Was any attempt made to understand the source of this bias and/or correct for it? Has this version of WAM (with the same wind-input and dissipation source terms) been validated in these conditions in other locations to show it is appropriate for simulating large wave heights? Are there similar biases in the wind field? These points should be commented on rather than trying to argue that this bias may be a good thing for some applications.*

To address this point some analysis of wind data was added at the end of this reply (Fig. 2R). It can be inferred that the wind also shows an overestimation compared with observations, in particular for high winds. Hence it appears likely that some if not most of the overestimation in wave heights (Fig.3R) may be attributed to the forcing. The wave model used in our study is used with minor modifications for forecasting at various weather services, such as for example the German Weather Service (DWD) where it shows that it is capable of producing reasonable wave forecast for mean and extreme conditions, provided that there is sufficient quality in the wind fields.
We tried to clarify and to better explain the sources of the bias in the revised manuscript.

*2. Please comment more on ERA-Interim data (Page 4 Line 5). Is the same wave model used? What data does it assimilate (is it the same data used in the verification, are errors consistent in the pre and post assimilation years)? What resolution was the ERA-Interim atmospheric forcing? Are the 6 hour outputs instantaneous or 6 hour mean?*

The ERA-Interim wave model is basically the same version of WAM as in our hindcast simulation. The ERA-Interim reanalysis uses not only satellite data to adjust the wave spectra since 1990, but also assimilate wind information from satellite and in situ wind observation since 1980, but do not include any observations from wave buoy data. The atmospheric part of the ERA-Interim reanalysis has a 30min time step and a spatial resolution of about 79km. The 6-hour values are instantaneous values, at 00, 06, 12 and 18 UTC. A detailed description is given in Dee et al. (2011). The information was also added to the revised manuscript.

*3. The wave results conclude in 2014, while the coastDat2 atmospheric results conclude in 2012. It should be explicitly mentioned if the time series was extended the additional years in a consistent manner.*

We clarified this issue in the manuscript and provide an explanation that the extension of the atmospheric forcing from 2012 to 2014 was done in a consistent way.

*Technical corrections*
*P1L16: 'Data from wind-wave hindcast data' <- Remove 2nd "data"* removed
*P2L11: 'based on' <- What does "based on" mean here?* clarified in text
*P2L13: 'Weisse and Günther (2007)' <- use parenthetical citation* changed
*P2L21: 'following-up' <- 'follow-up'* changed
*P3L6: 'only' <- Remove* removed
*P3L8: 'also' <- Remove* removed
*P3L10: Are these neutral winds? Is atmospheric stability accounted for?* clarified in text
*P3L13: '(Geyer, 2014)' <- Use in-line citation* changed
*P3L22: ', in addition,' <- Remove* removed
*P4L5: Remove 'also'* removed
*P4L8: 'onwards' <- onwards to what? 2017?* clarified in text
*Appendix A: The presentation here could be improved. E.g. in equation A1 isn't mean='\overbar{E}'? It may be clearer to replace this.*
*What is ME in equation A7?* clarified in text

*Appendix B: Some details about what the fields actually represent could be useful. E.g., How is significant wave height computed (and other parameters)? How is significant wave height (and other parameters) partitioned into wind-sea and swell components?*
A detailed description of how the wave parameters are derived is beyond the scope of this paper and can be found in the WAM documentation referred in the manuscript, or at https://github.com/mywave/WAM/blob/master/documentation/MyWave_D1.1.pdf.

*Figure 1: Add a figure title, e.g. 'North Sea Bathymetry' and include units label with colorbar.*

We added the units label but decided not to add a figure title as this is available from the caption.

*Figure 2: Add axes labels. Indicate in caption that these are the platform observations.*
We added the axes label and changed the caption.

*Figure 4: Please include x/y axes.*
We included a x/y axes to the figure.

*Figure 5:Please include x/y axes.*
We included a x/y axes to the figure.

*Figure 6: Add axes labels. Indicate in caption that these are the satellite observations.*
We added the axes label and changes the caption

*Figure 7: Add axes labels. Indicate in caption that these are the platform observations.*
We added the axes label and changes the caption

*Figure 8: Please include x/y axes.*
We included a x/y axes to the figure.

*Figure 9: Add axes labels. Indicate in caption that these are the platform observations.*
We added the axes label and changes the caption

*Figure 10: Add axes labels. A legend in 1 panel would make the figure easier to understand than describing in the captions.*
We added the axes label and a legend

[Figure]

*Figure 1R: Time series of yearly correlation for CD1 (red) and CD2 (cyan) and of Briers Skill score (magenta). Observational data at K13 are used as reference value for the Briers Skill Score.*

[Figure]

[Figure]

*Figure 2R: Comparison of 10m wind speed between observations (OBS), coastDat1(CD1) and coastDat2 (CD2) at the location K13 in the southern North Sea for the period 1979-2006. Top panel shows time series of 3h SWH for OBS (green), CD1 (red) and CD2(blue). Middle panels show scatter plot and Q-Q plot between OBS and CD1 (left) and OBS and CD2 (right). Lower panel shows box plots for 3 hour values for OBS, CD1 and CD2*

[Figure]

*Figure 3R: Comparison of the significant wave height for observations (OBS), coastDat1(CD1) and coastDat2 (CD2) at the location K13 in the southern North Sea for the period 1979-2006. Top panel shows time series of 3h SWH for OBS (green), CD1 (red) and CD2(blue). Middle panels show scatter plot and Q-Q plot between OBS and CD1 (left) and OBS and CD2 (right). Lower panel shows box plots for 3 hour values for OBS, CD1 and CD2*

[revised manuscript text omitted]

For the coarse grid simulation  monthly sea ice concentrations provided by the Hadley Centre Sea Ice and Sea Surface Temperature  dataset (HadISST1.1, Rayner et al., 2003) were used. The sea ice concentrations had a spatial resolution of 1° x 1°and were spatially interpolated to the coarse wave model grid. Subsequently in the simulation sea ice was accounted for by treating all model grid points with sea ice concentrations exceeding 50 % as land for the corresponding time steps.

**3.2  Reference data**

A number of in situ wind-wave observations from platforms and buoys in the North Sea originating from different sources were available for validation (Table 1). Basic quality control was applied but no attempt to check homogeneity was done. The

data cover different time periods including gaps and differ in their temporal resolution. Because the model output is available only at full model hours, the comparison with observations was limited to data that were measured within ±10 minutes around full hours.

In addition, wind-wave data derived from satellite provided by the merged altimeter wave height database version 11.0 (Queffeulou, 2013) were used. These data originate from the GlobWave project (http://www.globwave.org) and cover the period 1991 to present. The satellite data were co-located with the model data using a co-location criteria of ±10 minutes at which the position of the satellite was matched with the nearest grid point of the wave model.

To compare the wave model results in space and time  data from the ERA-Interim global reanalysis (Dee et al., 2011) spatially interpolated to the coastDat2 wave model grid were used. More specifically, data from the ocean wave product of ERA-Interim were used, which employs the same wave model as used in this study. ERA-Interim wave data are available every six hours at 00, 06, 12 and 18 UTC and at a spatial resolution of 0.75° latitude × 0.75° longitude.  Wind data from satellites and in situ observations are assimilated into ERA-Interim. Further, from 1990 onwards wave spectra  were adjusted using altimeter data . Wave buoy data were not used in the assimilation procedure (Dee et al., 2011). For comparison, we use the ERA-Interim wave data from 1980 to 2014.

[revised manuscript text omitted]

 The shorter and coarser ERA-Interim reanalysis  shows smaller errors for most locations and outperforms the coastDat2 hindcast in open waters. For coastal locations, for which unresolved bathymetry and land-sea distribution in ERA-Interim may play a role, improvements in coastDat2 are visible. Whereas the ERA-Interim wave product generally underestimates extreme significant wave heights, the coastDat2 hindcast has a tendency to overestimate. This is consistent with results from an analysis of marine wind fields used as forcing data which also showed some overestimation of the more severe wind speeds (Geyer et al., 2015). Compared to its predecessor the coastDat1 wave hindcast Weisse and Günther (2007), which terminates in 2007, the coastDat2 hindcast shows similar and comparable performance (not shown).

One benefit of the coastDat2 wind-wave hindcast  is the extended period for which reconstructed wave data are available  . In particular, the use for extreme value statistics or assessment of long-term variability and change require data sets as long as possible. Unfortunately, wave measurements to validate the hindcast over such extended periods of time are unavailable. There are, however, some indications that the data may be used for such analyses: The wave climate strongly depends on the wind climate and the quality and homogeneity of the wave hindcast is thus strongly coupled to those of the driving atmospheric data which are discussed in Geyer (2014) and in Geyer et al. (2015). In these papers, hindcast wind is compared with observations and for most locations, a slight overestimation and root mean square errors between $1.6\,\mathrm{ms}^{-1}$ and  $3.4\,\mathrm{ms}^{-1}$ were found. Moreover, the described long-term (decadal) variability, closely resembles the known variations in storm activity over the North Sea. The latter was shown to occur also for waves in a hindcast that uses wind fields from a different regional atmosphere model that was, however, driven by the same global reanalysis (Weisse and Günther, 2007).

Regarding homogeneity Kistler et al. (2001) investigated the issue for the global NCEP reanalysis that provided the boundaries for the regional atmosphere models used in this and the study of Weisse and Günther (2007). Kistler et al. (2001) showed that the skill for the Northern Hemisphere rapidly increased from 1948 to 1958 while there is only a small increase afterwards. The latter implies that the driving global reanalysis is mostly homogeneous for our area after about 1958 and, as there were no changes in regional model systems used in Geyer (2014) and our study, we conclude that the same may also hold for our wave hindcast.

[revised manuscript text omitted]